# Isolated Effects of Plasma Freezing versus Thawing on Metabolite Stability

**DOI:** 10.3390/metabo12111098

**Published:** 2022-11-11

**Authors:** Jane L. Buchanan, Jovan Tormes Vaquerano, Eric B. Taylor

**Affiliations:** 1Department of Molecular Physiology and Biophysics, University of Iowa Carver College of Medicine, Iowa City, IA 52240, USA; 2Holden Comprehensive Cancer Center, University of Iowa Carver College of Medicine, Iowa City, IA 52240, USA; 3Abboud Cardiovascular Research Center, University of Iowa Carver College of Medicine, Iowa City, IA 52240, USA; 4Pappajohn Biomedical Institute, University of Iowa Carver College of Medicine, Iowa City, IA 52240, USA; 5Fraternal Order of Eagles Diabetes Research Center (FOEDRC), University of Iowa Carver College of Medicine, Iowa City, IA 52240, USA; 6FOEDRC Metabolomics Core Research Facility, University of Iowa, Iowa City, IA 52240, USA

**Keywords:** metabolomics, freeze–thaw, plasma, LC–MS, metabolite stability

## Abstract

Freezing and thawing plasma samples is known to perturb metabolite stability. However, no study has systematically tested how different freezing and thawing methods affect plasma metabolite levels. The objective of this study was to isolate the effects of freezing from thawing on mouse plasma metabolite levels, by comparing a matrix of freezing and thawing conditions through 10 freeze–thaw cycles. We tested freezing with liquid nitrogen (LN_2_), at −80 °C, or at −20 °C, and thawing quickly in room temperature water or slowly on ice. Plasma samples were extracted and the relative abundance of 87 metabolites was obtained via liquid chromatography–mass spectrometry (LC–MS). Observed changes in metabolite abundance by treatment group correlated with the amount of time it took for samples to freeze or thaw. Thus, snap-freezing with LN_2_ and quick-thawing with water led to minimal changes in metabolite levels. Conversely, samples frozen at −20 °C exhibited the most changes in metabolite levels, likely because freezing required about 4 h, versus freezing instantaneously in LN_2_. Overall, our results show that plasma samples subjected to up to 10 cycles of LN_2_ snap-freezing with room temperature water quick-thawing exhibit remarkable metabolomic stability.

## 1. Introduction

Metabolite instability is a known source of variability in biological samples, particularly in blood and plasma samples that may undergo a variety of collection, storage, and handling procedures prior to metabolomic analysis. The effects of these pre-analytical variables on blood metabolites have been investigated by researchers as early as 2001, when Comstock et al. [1] examined how freeze–thaw cycles affect cholesterol and hormone levels in human plasma. Numerous studies have analyzed how blood collection tubes, hemolysis, and time and temperature of samples prior to metabolomics analysis alter metabolites in serum/plasma. However, studies differ widely by method, controls, and variables tested, making it difficult to directly compare one to another. For example, storage conditions can vary from 4 °C refrigerators to −80 °C freezers. Time in storage can vary from hours to weeks, months, or even years. Studies may utilize either plasma or serum, the latter taking longer to collect due to the time needed for clotting. Most studies have focused on human plasma/serum, although some have analyzed samples from mice [2] or rats [3]. The number of metabolites, type of metabolite, and how the metabolites are measured can also vary greatly depending on the study (Table 1). Studies have measured as few as 4 metabolites via chemical assays [4] and as many as 706 unconfirmed metabolites via untargeted metabolomics with external database matching [5]. 

Perhaps the variable that has been investigated most extensively is how the number of freeze–thaw cycles affects serum/plasma metabolite stability. Despite the abundance of freeze–thaw literature, no study has systematically isolated the effects of sample freezing from sample thawing and evaluated these effects separately on serum/plasma metabolite stability (Table 1). Therefore, the objective of this study was to utilize a novel design to isolate the effects of freezing from the effects of thawing on plasma metabolite stability.

## 2. Materials and Methods

### 2.1. Plasma Procurement and Handling

Blood was obtained via cardiac puncture from 5 6-month-old C57BL/6J female mice in accordance with the University of Iowa Institutional Animal Care and Use Committee (IACUC) guidelines (Protocol #0101249, Approved 1 July 2020). Approximately 700 µL of blood per mouse were added to 15 µL of EDTA (0.5 M) in 1.5 mL sterile microcentrifuge tubes on ice. Samples were mixed and centrifuged at 3000 rpm for 15 min at 4 °C. The plasma supernatant was extracted from each sample and pooled into a 15 mL polypropylene conical tube on ice. A total of 40 µL of pooled plasma was aliquoted into 1.5 mL sterile microcentrifuge tubes and immediately snap-frozen in liquid nitrogen (LN_2_). A total of 5 tubes were randomly selected from the LN_2_ and placed into a cardboard storage box that was filled with a thin layer of LN_2_ for a total of 5 boxes (one box/experimental condition). Each box was immediately moved to the same shelf of a −80 °C freezer for 24 h.

### 2.2. Freeze–Thaw Conditions

(1)*Control*: Samples remained in −80 °C storage for 10 days.(2)*Thaw in H_2_O, Freeze in LN_2_*: Samples were rapidly thawed in an RT (20 °C) water bath with gentle agitation until completely liquid (about 1 min). Samples were then snap-frozen in LN_2_ and returned to −80 °C storage. This procedure was performed each day for a total of 10 days (10 freeze–thaw cycles).(3)*Thaw in H_2_O, Freeze in −80 °C*: Samples were thawed in an RT water bath with gentle agitation until completely liquid (about 1 min). Samples were then returned to −80 °C storage (1–2 min until frozen). This procedure was performed each day for a total of 10 days (10 freeze–thaw cycles).(4)*Thaw in H_2_O, Freeze in −20 °C*: Samples were thawed in an RT water bath with gentle agitation until completely liquid (about 1 min). Samples were then transferred to −20 °C storage until completely frozen (about 4 h). After samples were frozen, they were moved to −80 °C storage. This procedure was performed each day for a total of 10 days (10 freeze–thaw cycles).(5)*Thaw on Ice, Freeze in LN_2_*: Samples were thawed on ice until completely liquid (~20 min). Samples were then snap-frozen in LN_2_ and returned to −80 °C storage. This procedure was performed each day for a total of 10 days (10 freeze–thaw cycles).

### 2.3. Metabolite Extraction

Extraction buffer was prepared by adding 2:2:1 methanol:acetonitrile:water to internal standards at 1 μg/mL each (D4-Citric Acid, ^13^C5-Glutamine, ^13^C5-Glutamic Acid, ^13^C6-Lysine, ^13^C5-Methionine, ^13^C3-Serine, D4-Succinic Acid, ^13^C11-Tryptophan, and D8-Valine; Cambridge Isotope Laboratories). Plasma volume was included in the calculations for the water portion of the buffer. A total of 720 µL of extraction buffer was then added to 40 μL of each plasma sample. Samples were placed on a rotating platform at –20 °C for 1 h and centrifuged at 4 °C for 10 min at 21,000× *g*. A total of 300 µL of the metabolite extracts was transferred into fresh microcentrifuge tubes. An equal volume of each extract was pooled to serve as a quality control (QC) sample, which was analyzed at the beginning, end, and after every tenth sample throughout the instrument run. Extraction buffer alone was analyzed as a processing blank sample to account for carryover or background contamination from the sample extraction process. Metabolite extracts, the QC sample, and the processing blank were evaporated to dryness using a speed-vacuum. All samples were reconstituted in 30 μL of acetonitrile/water (1:1, *v*/*v*), vortexed for 10 min, and incubated at –20 °C for 18 h. Samples were then centrifuged at 4 °C for 2 min at 21,000× *g* and the supernatant was transferred into autosampler vials for analysis.

### 2.4. LC–MS Analysis

For LC chromatographic separation, 2 µL of reconstituted metabolite extracts, QC sample, and processing blank were run on a Millipore SeQuant ZIC-pHILIC (2.1 × 150 mm, 5 µm particle size, Millipore Sigma #150460) column with a ZIC-pHILIC guard column (20 × 2.1 mm, Millipore Sigma #150437) attached to a Thermo Vanquish Flex UHPLC. The mobile phase comprised Buffer A (20 mM (NH_4_)_2_CO_3_, 0.1% NH_4_OH (*v*/*v*)) and Buffer B (acetonitrile). The chromatographic gradient was run at a flow rate of 0.150 mL/min as follows: 0–21 min-linear gradient from 80 to 20% Buffer B; 20–20.5 min-linear gradient from 20 to 80% Buffer B; and 20.5–28 min-hold at 80% Buffer B. Data were acquired using a Thermo Q Exactive Classic MS operated in full-scan, polarity-switching mode with a spray voltage set to 3.0 kV, the heated capillary held at 275 °C, and the HESI probe held at 350 °C. The sheath gas flow was set to 40 units, the auxiliary gas flow was set to 15 units, and the sweep gas flow was set to 1 unit. MS data acquisition was performed in a range of *m/z* 70–1000, with the resolution set at 70,000 (out of maximum 120,000), the AGC target at 10^6^, and the maximum injection time at 200 ms. The QC sample was analyzed at the beginning and end of the LC–MS run and after every 10 samples.

### 2.5. Mass Spectrometry and Data Analysis

Acquired LC–MS data were processed using the Thermo Scientific TraceFinder 5.1 software. Targeted metabolites were identified by matching accurate mass and retention times to the University of Iowa Metabolomics Core Facility’s in-house library of confirmed standards (Appendix A). During data analysis in TraceFinder, mass tolerance was set to 2 millimass units (0.002 Daltons). After peak area integration by TraceFinder, NOREVA software was applied for signal drift correction on a metabolite-to-metabolite basis using the pooled QC sample that had been analyzed throughout the instrument run as the normalizing reference [17]. We further accounted for technical variation in sample quantity and autosampler injection volume by normalizing each metabolite signal intensity to the sum of all metabolite signal intensities within a sample to generate a ratiometric metabolite fingerprint. NOREVA and ratiometrically normalized individual QC samples were superimposed on a PCA plot, providing evidence of high run quality and accurate data normalization (Appendix A). Internal standards were used to test for differences in extraction efficiency amongst samples, which were not observed. Batch effects were not present in this study since all samples were run in one batch on the LC–MS. The processing blank showed no evidence of carryover or contamination.

### 2.6. Quantification and Statistical Analysis

Using the NOREVA-corrected, ratiometric-normalized metabolite peak intensities, the fold-change between the treatment and control samples for each metabolite was calculated and analyzed by unpaired Student’s *t*-test. P-values were negative log10-transformed, and fold-change data were log2-transformed, averaged, and plotted in volcano plots. Fold-change data were also analyzed by one-way ANOVA with the Holm–Sidak post hoc multiple comparison test compared to control samples. Statistically significant differences were defined as having *p*-values < 0.05 (*), *p* < 0.01 (**), *p* < 0.001 (***), and *p* < 0.0001 (****). Results were plotted in bar graphs as mean ± standard error relative to the control samples. The number of technical replicates used in each experiment is specified in the relevant figure legend. NOREVA-corrected, ratiometrically normalized metabolite peak intensities were visualized by PCA using MetaboAnalyst [18] and ClustVis software [19]. BioRender.com was used to create the experimental schematic in Figure 1A. Statistics, Figure 1C–F, and Figure 2 were generated using GraphPad Prism 9.4.1.

## 3. Results

### 3.1. General Trends

A total of 87 metabolites were identified in all plasma samples by liquid chromatography–mass spectrometry (LC–MS). The treatment groups were separated via principal component analysis (PCA) (Figure 1B), with samples frozen in −20 °C exhibiting the greatest change in the measured metabolome. Fatty acids tended to increase in all freeze–thaw conditions (Figure 2A), in addition to dihydrobiopterin and ornithine (Figure 2B), consistent with the release of these metabolites from albumin or other proteinaceous plasma components [20,21]. Conversely, the adenosine levels decreased with all types of freeze–thawing, suggesting sensitivity to degradation with phase changes (Figure 2B).

### 3.2. Effects of 10 Freeze–Thaw Cycles (LN_2_ Snap-Freeze, RT Water Rapid Thaw) vs. Chronic −80 °C Storage

Interestingly, there were few statistically significant (*p* < 0.05) metabolite differences between the samples that were stored in a −80 °C freezer after a single snap-freeze with RT water rapid-thaw cycle (control) and samples that underwent 10 freeze–thaw cycles (LN_2_ snap-freeze, RT water rapid thaw) (Figure 1C). Consistent with general trends, metabolites that significantly increased in fold change with multiple freeze–thaws were predominantly fatty acids, such as myristic, palmitic, lauric, and pentadecanoic acid. Adenosine was the only metabolite to significantly decrease. 

### 3.3. Effects of Thawing on Ice vs. Water (10 Cycles, LN_2_ Snap-Freeze)

Samples that were thawed on ice instead of RT water showed metabolomic changes comparable to samples frozen in −20 °C. Cystine and adenosine significantly decreased, while ornithine, allothreonine, and dihydrobiopterin significantly increased (Figure 1D).

### 3.4. Effects of Freezing in −20 °C vs. LN_2_ (10 Cycles, RT Water Rapid Thaw)

The samples that were frozen in −20 °C underwent the most metabolomic change of any treatment group compared to the samples that were snap-frozen in LN_2_ (10 cycles each, RT water rapid thaw for all), with five metabolites significantly decreasing and five significantly increasing. Notably, the disulfide-containing compounds (cystine, oxidized glutathione) decreased, and the amino acids or amino acid-derivatives (isoleucine, ornithine, allothreonine) significantly increased (Figure 1E). Similar to the samples that were frozen in −80 °C, adenosine significantly decreased, and dihydrobiopterin significantly increased.

### 3.5. Effects of Freezing in −80 °C vs. LN_2_ (10 Cycles, RT Water Rapid Thaw)

Metabolites from the samples that were frozen in −80 °C showed few changes compared to the samples that were snap-frozen in LN_2_ (10 cycles each, RT water rapid thaw for all). Adenosine was the only metabolite to significantly decrease, while three metabolites (dihydrobiopterin, tyrosine, suberic acid) significantly increased (Figure 1F).

## 4. Discussion

To the best of our knowledge, no study examining the stability of the blood or plasma metabolome through freeze–thaw cycles has systematically isolated the effects of sample freezing from sample thawing. By testing three freezing conditions (LN_2_: instantaneous; −80 °C: 1–2 min; −20 °C: ~4 h) while keeping the thawing condition constant (H_2_O: 1–2 min), we were able to determine how the method of freezing impacted the metabolome. By testing two thawing conditions (H_2_O: 1–2 min, ice: ~20 min.), while keeping the freezing condition constant (LN_2_: instant), we were able to compare how the method of thawing impacted the metabolome.

The samples that were cycled 10 times with freezing at −20 °C showed the most metabolomic changes, including decreases in adenosine and disulfide compounds and increases in ornithine, isoleucine, and dihydrobiopterin. Conversely, the metabolites from the samples cycled 10 times with freezing at −80 °C or LN_2_ showed few changes. Interestingly, thawing on ice led to many of the same metabolomic changes as freezing at −20 °C. These results suggest that increased time in the liquid–ice transition may lead to increased metabolite degradation or an increase in other metabolites due to the desorption from protein. It is known that protein denaturation can occur in buffered solutions at the ice–liquid interface where the pH changes are the greatest [22,23] and that protein aggregation can increase with slow freezing (>4 h) [24]. Consequently, adenosine and disulfide compounds may undergo degradation with increased freezing or thawing duration due to increased exposure to pH changes or molecular crowding that occurs with protein aggregation.

Several studies have also shown that a decrease in pH can lead to the dissociation of fatty acids from albumin and to an increased binding to phospholipid bilayers from cell membranes [20,21]. Interestingly, we observed an increase in fatty acid levels in all samples that underwent 10 freeze–thaw cycles (LN_2_ freeze, RT water rapid thaw) compared to the samples that remained in −80 °C storage (freeze–thawed once before processing). This suggests that fatty acids may be dissociated from albumin or other proteinaceous plasma components during phase changes associated with freeze–thawing. It is important to note that trace fatty acid contamination is a challenge associated with metabolomics studies, and it is possible that our samples could have been exposed to fatty acids in the environment. However, we believe that it is unlikely that all sample groups would have become contaminated with fatty acids, since our sample microcentrifuge tubes remained tightly shut throughout the freeze–thaw procedures. Additionally, both the sample and control tubes received the same amount of environmental exposure during metabolite extraction and metabolomics sample processing. Because changes in pH can cause the dissociation of fatty acids from albumin, and freeze–thawing is accompanied by pH changes, we believe that the simplest interpretation of the marginal increase in fatty acids we observe is as a bona fide versus artifactual effect.

There are notable similarities and differences between this study and the freeze–thaw studies listed in Table 1. However, substantial variation in design prevents systematic metabolite-by-metabolite comparison of most of them to the present study. For instance, several studies focused on specific metabolites, such as Alzheimer’s biomarkers (e.g., neurofilament light chain, tau) (4) or hormones (e.g., testosterone, progesterone) [1]. Many studies used chemical assays or NMR to detect metabolites and reported less than 40 metabolites. The studies that used mass spectrometry and reported more than 700 metabolites either did not confirm metabolite identity [5] or measured differences in the total ion chromatograph after freeze–thawing and did not report specific metabolites [16]. Chen et al. [7], Brier et al. [10], and Anton et al. [9] are most comparable to this study in that mass spectrometry was used to measure more than 40 metabolites that are routinely found in plasma. Chen et al. compared the effects of 2, 3, 4, or 5 freeze–thaw cycles on individual human plasma samples to 1 freeze–thaw cycle and found no specific freeze–thaw cycle markers. Brier et al. collected human serum and plasma on three different days and found that all 159 metabolites maintained stable concentrations after one freeze–thaw cycle, with the exception of methionine sulfoxide, a methionine derivative. A total of 11 out of 159 metabolites revealed significantly decreased concentrations after two freeze–thaw cycles, including decenoic acid (C10:1), three amino acids (isoleucine, tryptophan, and valine), five phosphatidylcholines, and acetylornithine. In contrast, we found that isoleucine and ornithine significantly increased with freezing at −20 °C, although the fold-changes were small. Decenoic acid and phosphatidylcholines are not part of our in-house library of standards and, therefore, were not measured. Notably, Anton et al. also found that isoleucine and ornithine significantly increased, but only during 12, 24, and 36 h storage at room temperature and not during freeze–thaw cycles. Anton et al. reported that no significant metabolite concentration changes occurred for up to four freeze–thaw cycles. Overall, a consistent theme that emerges from Chen et al., Brier et al., Anton et al., and our own study is that most of the measured metabolites do not change significantly after multiple freeze–thaw cycles.

A limitation of this study is that our LC–MS analysis was constrained to 87 metabolites. Although the number of standards in our LC–MS library is closer to 400 metabolites, the 87 targeted metabolites (Appendix A) are representative of the metabolites that we routinely detect in plasma without a more complex workflow of running different concentrations of extracts. Notably, this is a significantly higher number of detected metabolites than most freeze–thaw studies report. The targeted metabolites in this study represent a wide variety of compound classes, most of which were resistant to freeze–thaw perturbations. It is possible that the low abundance metabolites that were not detected could be more susceptible to damage from freeze–thaw cycling. Nonetheless, we expect that the stability patterns we observed with the metabolites that we detected likely apply to metabolites that we did not detect. Another limitation of this study is that we used mouse plasma, which is not a perfect metabolomic replica of human or other animal plasma. However, because the general composition of plasma across most animals is similar, we expect our findings with mouse plasma will be generally conserved to human plasma.

It is important to note that collecting human serum/plasma samples presents numerous challenges that collecting blood from animals in a laboratory setting may not. For instance, serum/plasma samples from patients may not be stored immediately and clinics may not have access to LN_2_ for rapid sample freezing. Patient serum/plasma samples may also undergo multiple freeze–thaw cycles if they are stored in more than one location or are used for multiple studies. However, because our study indicates which metabolites are sensitive to degradation in specific freezing and thawing conditions, researchers can retrospectively apply that information to their plasma metabolomics studies. For instance, a recently published study by our collaborators illustrates the plasma metabolomic changes that can occur in diabetic patients with chronic kidney disease [25]. Sun et al. highlights orotate and citrate as metabolites that change significantly when diabetic patients are treated with the drug allopurinol. Since we show that orotate and citrate do not change significantly with any freeze or thaw condition for up to 10 freeze–thaw cycles, the observations in Sun et al.’s patient data are likely not due to an artifact of plasma metabolomics sample handling.

## 5. Conclusions

By isolating the effects of freezing from the effects of thawing on the levels of 87 plasma metabolites, we show that an increased duration of freezing or thawing results in an increased number of metabolite-level changes, suggesting that rapid freezing by LN_2_ and rapid thawing in water are the best practices for metabolomics studies involving plasma samples. Additionally, we showed that plasma samples that undergo up to 10 freeze–thaw cycles with rapid freezing and rapid thawing exhibit remarkably stable metabolite levels. These results will help to inform future workflows for preparing plasma samples for metabolomics analyses.

## Figures and Tables

**Figure 1 metabolites-12-01098-f001:**
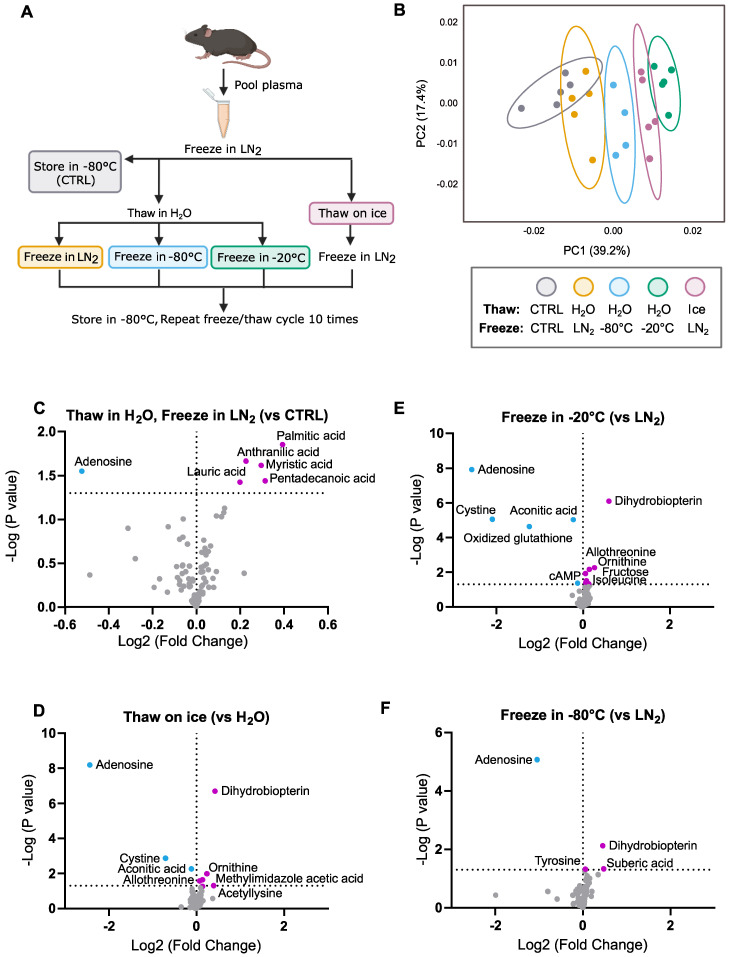
(**A**) Schematic of the freeze–thaw experimental design. (**B**) Principal component analysis (PCA) of relative metabolite levels for each freeze–thaw condition (10 cycles each condition). *Isolation of freeze–thaw cycling*: (**C**) Volcano plots showing the average log2 fold change of metabolites from samples that underwent 10 freeze–thaw cycles (LN_2_ snap-freeze, RT water rapid thaw for all) vs. samples that were stored in −80 °C and underwent 1 freeze–thaw (CTRL). *Isolation of thaw variable:* (**D**) Volcano plots showing the average log2 fold change of metabolites from samples thawed on ice compared to samples rapidly thawed in RT water (LN_2_ snap-freeze for each). *Isolation of freeze variable:* Volcano plots showing the average log2 fold change of metabolites from (**E**) samples frozen in −20 °C or (**F**) −80 °C relative to samples snap-frozen in LN_2_ (RT water rapid thaw for each). *p*-values were calculated by Student’s *t*-test (*n* = 4–5 technical replicates). Blue dots denote a statistically significant (*p* < 0.05) decrease in log2 fold change. Pink dots denote a statistically significant increase in log2 fold change. Dotted *x*-axis indicates *p* = 0.05.

**Figure 2 metabolites-12-01098-f002:**
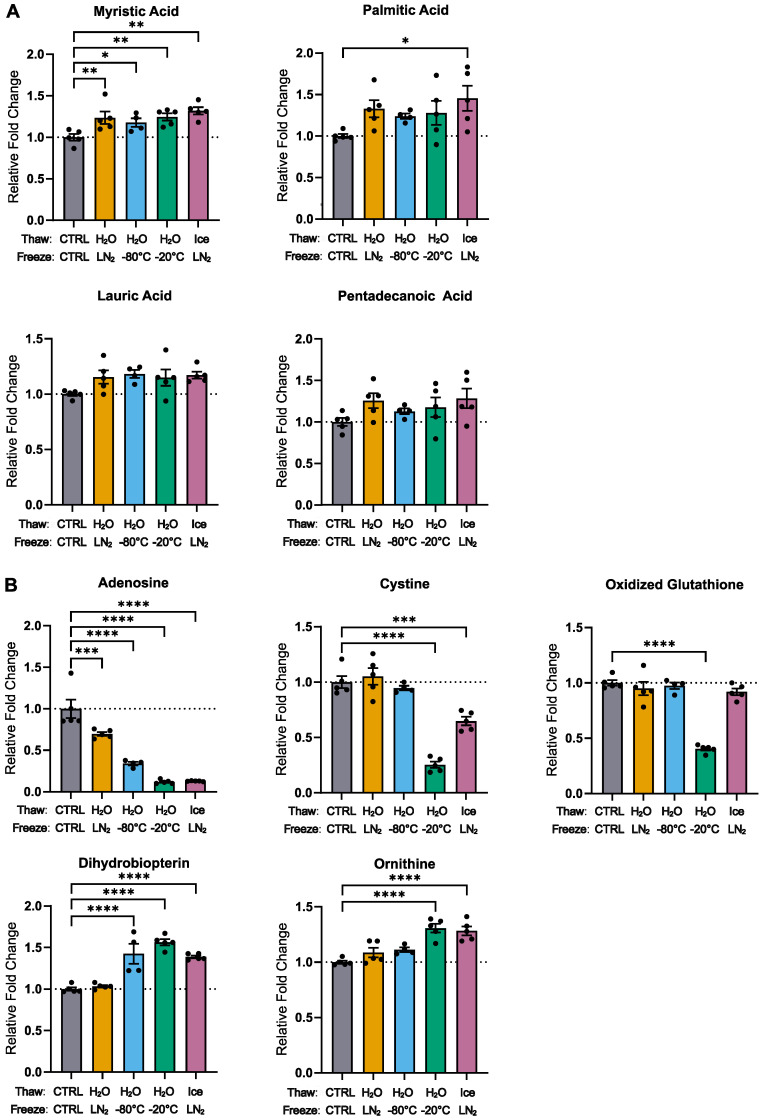
Bar graphs comparing the most changed metabolites across freeze and thaw conditions (10 cycles each) except for the control (CTRL) that was cycled 1 time. (**A**) Fatty acids increase in all freeze–thaw conditions compared to control. (**B**) Specific metabolites decrease or increase depending on freeze–thaw condition. Data are presented as the mean ± standard error. *p*-values were calculated by one-way ANOVA with the Holm–Sidak post hoc multiple comparison test compared to control; * *p* < 0.05, ** *p* < 0.01, *** *p* < 0.001, **** *p* < 0.0001. (*n* = 4–5 technical replicates).

**Table 1 metabolites-12-01098-t001:** Plasma/serum freeze–thaw studies that isolate freezing variables (orange), isolate thawing variables (blue), or do not isolate different freezing or thawing variables (pink).

Study Citation	Analytical Method	of Peaks/Metabolites Analyzed	of Freeze–Thaw Cycles (1 = Control)	Thawing in Water			Freezing in LN_2_		Varied Thawing and Freezing			
				Water Thaw, LN_2_ Freeze	Water Thaw, −20 °C Freeze	Water Thaw, −80 °C Freeze	Ice Thaw, LN_2_ Freeze	RT Thaw, LN_2_ Freeze	Ice Thaw, −80 °C Freeze	RT Thaw, −20 °C Freeze	RT Thaw, −40 °C Freeze	RT Thaw, −80 °C Freeze
This study	LC–MS	87	1, 10	X	X	X	X					
An et al. 2021 [6]	iTRAQ-LC–MS/MS	30	1, 2, 3, 4						X			
Chen et al. 2020 [7]	LC–MS/MS	150	1, 2, 3, 4, 5									X
Wang et al. 2019 [8]	NMR	20	1, 2, 3, 4, 5									X
Keshavan et al. 2018 [4]	Assay	4	1, 2, 3, 4									X
Torell et al. 2017 [2]	GC-TOF-MS	46	1, 2						X			
Anton et al. 2015 [9]	FIA-ESI-MS/MS	127	1, 2, 3, 4	X			X	X				X
Breier et al. 2014 [10]	ESI-LC–MS/MS	188	1, 2, 3		**?** ^1^					**?** ^1^		
Pinto et al. 2014 [11]	NMR	8 *	1, 2, 3, 4, 5									X
Cuhadar et al. 2013 [12]	Assay	17	1, 2, 3, 4, 5, 6, 7, 8, 9, 10							X		
Yin et al. 2013 [5]	UPLC-qTOF-MS	706	1, 2, 4			X						
Fliniaux et al. 2011 [13]	NMR	8 *	1, 5, 10									X
Wood et al. 2008 [14]	LC–MS	15	1, 2, 3, 4			**?** ^2^						**?** ^2^
Teahan et al. 2006 [15]	NMR	10 *	0, 1 ^3^								X	
Mitchell et al. 2005 [16]	C-MALDI-TOF-MS	2153 **	1, 2, 3, 4, 5									X ^4^
Deprez et al. 2002 [3]	NMR	30 *	0, 1 ^3^					X				
Comstock et al. 2001 [1]	Assay	22	1, 2, 3, 4, 6, 10			X ^4^						

^1^ Samples were frozen at −20 °C, but how samples were thawed was not specified. ^2^ Samples were frozen at −80 °C, but how samples were thawed was not specified. ^3^ One freeze–thaw cycle was compared to no freeze–thaw cycles. Other studies compare more than one freeze–thaw cycle to one freeze–thaw cycle. ^4^ Samples were frozen at −70 °C. * Metabolites were named in figures or tables, but the number of measured metabolites was not reported. ** Specific metabolites were not identified. RT Thaw = samples were thawed in room temperature air or unspecified.

## Data Availability

Data supporting the reported results can be found in the Appendix A.

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
