# Peer review of "Isolated Effects of Plasma Freezing versus Thawing on Metabolite Stability"

_metabolites, 2022, doi:10.3390/metabo12111098_

Round 1

Reviewer 1 Report

The manuscript submitted for publication in Metabolites refers on the effects of plasma freezing/thawing conditions on the metabolites stability. Liquid nitrogen, -80 oC and -20 oC were considered as alternative freezing conditions, while room temperature water fast process and -20 oC slow process were considered as alternative thawing conditions. Ten consecutive freezing/thaw cycles have been investigated.

The manuscript is well written. Design of experiments was accurately established and clearly explained. Results were correctly interpreted and discussed. The findings of the study are interesting and the manuscript will induce interest of the scientific community.

Consequently, I recommend publication of the manuscript in the presented form.  

Reviewer 2 Report

The study carried out a systematic evaluation of the effects of freezing and thawing methods and freeze-thaw cycles on selected metabolites. Although the research is of interest, there is a number of issues that need clarifications.

Major concerns.

Because the study performed semi-quantitative analysis, one would question the robustness and reliability of the sample analysis, for example, internal standard recoveries across the samples, QC sample results, steps taken to mitigate batch effects, carryovers, and technical variations. In terms of data analysis, crucial information such as library matching criteria and blank correction is missing. All of these can help validate the outcomes of the study.

Line 61-62: The results demonstrate increases in some fatty acids with respect to freeze-thaw cycles according to Figure 2A. Among the selected fatty acids, only myristic acid exhibited significant changes between each freeze-thaw condition and control. All fatty acids show relative fold change < 1.5. The small fold changes and lack of statistical significance for some cases leads to the question as to whether the increases could fall under technical variations. From our experience and literature, lab contamination could introduce traces of fatty acids, especially palmitic acid. Therefore, going back to my first concern, it is necessary to provide compelling evidence supporting the integrity of the analytical method.  

Discussion: I found the discussion is rather short which could have been improved. For example, it is mentioned only 89 metabolites were targeted (which I could not find information in the main manuscript about what classes of compounds were these) but there is no discussion on other classes that seem robust to freeze-thaw or how the findings are compared to other studies. The authors have brought up this point in the introduction so I hope to see more discussion or suggestion on this.  

Minor comments.

Table 1:

-          I am not sure how to extract information from the studies with N/A number of features/metabolites analyzed. Also, it would be interesting to see what classes of metabolites identified in those studies.

Table 2

-          Information on the second row is redundant. It is already given in the caption.

-          The fact that the pink group did not either test freezing or thawing variables led me to question the purpose of including this in the comparison. Maybe the provided description is not clear on what kind of information we could get from the studies.

-          Please consider combining Table 1 and 2 as a single table.

-          In addition to information on who did what, a brief finding could be added to each study.

Results

Line 59-60. The authors report the separation among treatments based on PCA (Figure 1B) which shows five clusters. I wonder if the clustering is arbitrary or what algorithm was used. Also, it might be better to present the PCA (or a separate PCA) with QC samples so that readers can evaluate the robustness of the sample analysis.

Line 71: Should it be Figure 1F? Please carefully check if Figure 1C-F are correctly referenced throughout the text. I feel that some information in the caption and in the text are mismatched or maybe it is not clearly written.

Line 88: should it be Figure 1D?

Sections 2.2-2.4: What is the log2 (fold change) cut-off for significant metabolites? Some of them are very close to the middle which appear not to contribute much to the differences. For example, ornithine, fructose, isoleucine, cAMP and allothronine in Figure 1D.

Line 212: RT for room-temperature

Line 216: remove “(RT)”

Line 235: replace uL with µL

Line 265: what were the mass and retention time tolerances used for metabolite identification?

Line 269: If internal standards were not chosen to normalize metabolite intensities, how were technical variations accounted for?

All Figures: replace with high-resolution figures.

Reviewer 3 Report

Dear editor;

Regarding the article Number metabolites-1975424; titled Isolated effects of plasma freezing versus thawing on metabolite stability

I am attaching my comments for your concern as follows;-

-       The article is well designed

Title: good.

Abstract:

-       I did not see the objective in the abstract, authors should add it.

-       Also, conclusion in the abstract should be revised and rewritten

Introduction:

-       Well written, the objective at the end of the introduction need to be clarified and restate.

Materials & methods

-       Where is the ethical approval for this study.

-       In section 4.2, Freeze-Thaw Conditions, no need for the underlines (1-4), please remove it.

Results

-       Tables 1 and 2 needed to be redesigned according to the journal instructions.

-       Figure 1. (A) Schematic of the freeze-thaw experimental design need to be redesigned for more clarification.

-       Are there any Signiant differences in the amount of the compared isolated at different conditions?

-       Author need to compare it statistically.

Discussion:

-       Authors need to discuss more about the current results and compared it with the previous one or similar studies.

Round 2

Reviewer 2 Report

The responses are accepted.